Global warming and neurological practice: systematic review

Amiri Moshgan 1
Peinkhofer Costanza 1 2
Othman Marwan H. 1
De Vecchi Teodoro 1 2
Nersesjan Vardan 1
Kondziella Daniel daniel_kondziella@yahoo.com 1 3
1 Department of Neurology, Rigshospitalet, Copenhagen University Hospital , Copenhagen , Denmark
2 Medical Faculty, University of Trieste , Trieste , Italy
3 Department of Clinical Medicine, Faculty of Health and Medical Sciences, University of Copenhagen , Copenhagen , Denmark
Abdullah Jafri
Electronic publication date: 2021 Aug 4
Publication date: 2021
Volume: 9
Electronic Location ID: e11941
Received 2021 Apr 28; Accepted 2021 Jul 19
Copyright: ©2021 Amiri et al.
Copyright year: 2021
Copyright holder: Amiri et al.
License: This is an open access article distributed under the terms of the Creative Commons Attribution License, which permits unrestricted use, distribution, reproduction and adaptation in any medium and for any purpose provided that it is properly attributed. For attribution, the original author(s), title, publication source (PeerJ) and either DOI or URL of the article must be cited.
License URL: https://creativecommons.org/licenses/by/4.0/

Keywords: Neurology, Multiple sclerosis, Alzheimer dementia, Migraine, Epilepsy, Stroke, Global warming, Climate change, Migration, Epidemiology

Funding: The Lundbeck Foundation R349-2020-658 This work was supported by the Lundbeck Foundation (R349-2020-658). The funders had no role in study design, data collection and analysis, decision to publish, or preparation of the manuscript.

==============================
Background

Climate change, including global warming, will cause poorer global health and rising numbers of environmental refugees. As neurological disorders account for a major share of morbidity and mortality worldwide, global warming is also destined to alter neurological practice; however, to what extent and by which mechanisms is unknown. We aimed to collect information about the effects of ambient temperatures and human migration on the epidemiology and clinical manifestations of neurological disorders.

Methods

We searched PubMed and Scopus from 01/2000 to 12/2020 for human studies addressing the influence of ambient temperatures and human migration on Alzheimer’s and non-Alzheimer’s dementia, epilepsy, headache/migraine, multiple sclerosis, Parkinson’s disease, stroke, and tick-borne encephalitis (a model disease for neuroinfections). The protocol was pre-registered with PROSPERO (2020 CRD42020147543).

Results

Ninety-three studies met inclusion criteria, 84 of which reported on ambient temperatures and nine on migration. Overall, most temperature studies suggested a relationship between increasing temperatures and higher mortality and/or morbidity, whereas results were more ambiguous for migration studies. However, we were unable to identify a single adequately designed study addressing how global warming and human migration will change neurological practice. Still, extracted data indicated multiple ways by which these aspects might alter neurological morbidity and mortality soon.

Conclusion

Significant heterogeneity exists across studies with respect to methodology, outcome measures, confounders and study design, including lack of data from low-income countries, but the evidence so far suggests that climate change will affect the practice of all major neurological disorders in the near future. Adequately designed studies to address this issue are urgently needed, requiring concerted efforts from the entire neurological community.

Introduction

The United Nations has identified climate change, which includes global warming, as the “defining issue of our time” (Climate Change, 2020). As stated in the Paris Climate Agreement, the increase of global temperature must be contained within 2, ideally 1.5, degrees Celsius above pre-industrial levels (Paris agreement, United Nations, 2015) to avoid major negative consequences, including poorer global health and a rise of environmental refugees (i.e., displaced people from increasingly inhabitable regions of the world). Thus, climate change and global warming will likely worsen illnesses worldwide (Climate Change Impacts Human Health, 2020) and may become the major drivers of human migration (Climate Change, 2020).

According to the Global Burden of Disease Study, neurological disorders are the foremost cause of disability adjusted life years (DALYs), accounting for 10.2 percent of global DALYs, and neurological diseases are the second-leading cause of deaths, representing 16.8 percent of global deaths (GBD, 2015). The effects of environmental factors on neurological practice are multifaceted and complex, involving climate, temperature, biodiversity, toxins, food availability, quality and nutritional status and differences in human habitation and many others (Reis et al., 2021). It is therefore reasonable to assume that climate change and global warming will also have a major impact on clinical neurological practice.

To start addressing this issue, one must first identify the areas of neurological practice that will likely be subject to alterations related to global warming. It is then necessary to investigate in which ways a rise of ambient temperatures might affect the frequency, semiology, and outcome of major neurological disorders. Furthermore, the prevalence and incidence of neurological disorders in refugees and other human migrant populations might serve as a proxy for what could be expected in future environmental refugees.

In this review, our main objective was to identify how ambient temperatures influence the epidemiology and clinical aspects of major groups of neurological disorders. The second objective was to investigate the prevalence and incidence of neurological disorders in human refugee and migrant populations.

Survey methodology

Using the PICO approach (Schardt et al., 2007), we phrased the following research questions:

1. In people with major neurological disorders, including Alzheimer’s and non-Alzheimer’s dementia (AD and non-AD), epilepsy, headache and migraine, multiple sclerosis (MS), Parkinson’s disease (PD), stroke, and tick-borne encephalitis (TBE) (as a proxy for neuroinfectious diseases) (Population), how does an increase in ambient temperatures (Intervention), as compared to normal ambient temperatures (Comparison), affect the frequency, symptomatology, and mortality of these disorders (Outcome)?

2. In human refugee and migrant populations (Population), how does the fact of being displaced (Intervention), as compared to populations living in their home country (Comparison), affect the prevalence and incidence of neurological disorders (Outcome)?

For each PICO question a systematic review of the literature was performed using a predefined search. The full search strategies (including MeSH headings for searches in PubMed) for each of these two PICO questions, as well as the detailed search strings, are available from the Supplemental Information. The review was pre-registered at PROSPERO (2020 CRD42020147543).

Types of studies, and inclusion/exclusion criteria

Inclusion criteria

a. Studies published in English and listed in MEDLINE (PubMed) and Embase from January 1, 2000.

b. Non-English publications if an English abstract and a reliable translation of the manuscript into English were available

c. Cross-sectional or longitudinal, retrospective or prospective, observational clinical and research studies presenting original data about major neurological disorders (Alzheimer’s disease, stroke, epilepsy, neuroinfectious diseases, Parkinson’s disease and multiple sclerosis)—morbidity, frequency and/or mortality or any subtype—in response to an increase to ambient temperature were included.

d. Cross-sectional or longitudinal, retrospective or prospective, observational clinical and research studies presenting original data about major neurological disorders (headache/migraine, Alzheimer’s disease, stroke, epilepsy, neuroinfectious diseases, Parkinson’s disease or multiple sclerosis) in refugee and/or migrant populations.

Exclusion criteria

a. Studies published prior to January 1, 2000.

b. We excluded commentaries, summaries, reviews, editorials, animal studies, toxicological studies, duplicates, and studies that reported another association not related to our objectives.

c. In cases of missing data (those we cannot extract), the authors were contacted for additional information. If no answer was obtained, the study was excluded.

Inclusion criteria were human studies, published in English (or when a reliable English translation was available) and listed in MEDLINE (PubMed) or Scopus from January 1st, 2000 onwards; we included cross-sectional or longitudinal, retrospective or prospective, observational clinical and research studies presenting original epidemiological and/or clinical data about major neurological disorders as outlined above and their morbidity and/or mortality and/or prevalence/incidence, in relation to an increase in ambient temperature (any), respectively, refugee and/or migrant populations (irrespective of whether comparisons were made between migration groups with populations from the place of origin or place of departure).

Exclusion criteria were publications prior to January 1st, 2000; lack of a reliable English translation; and we excluded commentaries, summaries, reviews, editorials, animal studies, toxicological studies, and duplicate studies; and in cases of missing data, the authors were contacted for additional information. If no answer was obtained, studies were excluded as well.

Electronic search strategy

We included human only studies published in English and listed in PubMed and Scopus from January 1, 2000 to December 31st, 2020. Non-English literature was included if an English abstract and a reliable translation of the manuscript into English were available. The literature search was supervised by an information specialist from the Copenhagen University Library Service, and the search strategies were developed in accordance with the above PICO questions. For search strings and filters, the reader is referred to the Supplemental Information. The references of relevant articles were manually searched to identify additional articles. Further, papers were cross-referenced using the ‘cited by’ function in PubMed. When necessary, personal communication with authors was attempted by email or phone to obtain additional data.

Data collection

After reviewing titles and abstracts, relevant studies were assessed on a full text basis. Data were extracted in duplicate by MA, CP, MHO and cross-checked by DK; any uncertainties concerning data extraction and interpretation were resolved by consensus and the senior author DK.

Statistical analyses and reporting of bias

Owing to the high heterogeneity of the data including different definitions of temperature thresholds, quantitative statistical analysis was not performed. For the same reasons, and for the expected lack of adequately designed studies to meet the topic of this review, i.e., how climate change will alter neurological practice, we decided against using formal bias assessment.

Results

The primary searches yielded 4,996 titles in total. After screening and removal of duplicates, 93 studies met the inclusion criteria for the final review (Fig. 1). Table 1 provides an overview of studies (n = 84) regarding ambient temperatures and major neurological disorders. The effects of migration on prevalence and incidence of neurological disorders (n = 9 studies) are outlined in Table 2.

Figure 1 Flow chart.

This figure depicts a flow chart diagram of the literature search.

Ambient temperature and major neurological disorders

Alzheimer’s and non-Alzheimer’s dementia

Eleven studies investigated the effects of high ambient temperatures on patients with Alzheimer’s and non-Alzheimer’s dementia (Cornali et al., 2004; Hansen et al., 2008; Page et al., 2012; Zanobetti et al., 2013; Trang et al., 2016; Culqui et al., 2017; Conti et al., 2007; Lee et al., 2018; Linares et al., 2017; Zhang et al., 2016b; Xu et al., 2019). The studies were from eight countries with a total case population of N = 961,582. One study did not specify the number of cases (Hansen et al., 2008).

Maximum temperatures reported were >30 °C in nine studies (Cornali et al., 2004; Hansen et al., 2008; Zanobetti et al., 2013; Trang et al., 2016; Culqui et al., 2017; Conti et al., 2007; Linares et al., 2017; Zhang et al., 2016b; Xu et al., 2019), and respectively 18 °C and >24.5 °C in two of the studies (Page et al., 2012; Lee et al., 2018). In nine of the 11 studies, high ambient temperatures were associated with worsening of symptoms including agitation, as well as an increase in hospitalizations and/or mortality (Cornali et al., 2004; Hansen et al., 2008; Zanobetti et al., 2013; Culqui et al., 2017; Conti et al., 2007; Lee et al., 2018; Linares et al., 2017; Zhang et al., 2016b; Xu et al., 2019). Five studies suggested that high temperature was attributable to worsening of symptoms and increased risk of admission (Cornali et al., 2004; Culqui et al., 2017; Lee et al., 2018; Linares et al., 2017; Zhang et al., 2016b) while Hansen et al. found a significantly increased rate of both hospital admissions and mortality in 94,447 patients with dementia during periods of heat waves (Hansen et al., 2008). Zanobetti et al., (2013), Conti et al. (2007) and Xu et al. (2019) also found significantly increased mortality during hot months. In the remaining two studies, no associations were found between high ambient temperatures and hospital admissions or death in people with dementia (Page et al., 2012; Trang et al., 2016). Overall, a negative impact of increased temperatures on hospitalization and mortality in patients with dementia was reported in nine of the 11 studies.

Table 1 Characteristics of studies included in the review, investigating the association between ambient temperatures and major neurological disorders.

Temperatures are given as maximum temperatures or increase in mean temperature. Only the effects of high temperature on epidemiology, symptoms and mortality are listed.

Diagnosis	Number of studies (%)*	Study design	Site of study	Total case population	Temperature (maximal)	Effect on epidemiology (increase in incidence/ prevalence/ hospitalization)	Worsening of symptoms	Mortality (increase)	
Stroke	58 (69.0%)	Retrospective, observational	AU, CA, CN, DE, DK, ES, GB, IN, IL, IT, JP, KR, PR, QA, RU, SE, TU, TW, US	5,869,284	12.9 °C in 1 study 20–27 °C in 12 studies > 30 °C in 16 studies N/A in 5 studies	17 studies	N/A	17 studies	
Alzheimer and non-Alzheimer dementia	11 (13.1%)	Retrospective, observational	AU, CN, ES, IT, KR, UK, US, VN	961,582	18 °C in 1 study > 24.5 °C in 1 study >30 °C in 9 studies	6 studies	2 studies	4 studies	
Multiple Sclerosis	6 (7.1%)	5 studies: Retrospective, observational 1 study: Retro- and prospective, observational	AU, DE, FR, JP, US	5,305	>20 °C	N/A	4 studies	N/A	
Headache and Migraine	2 (2.3%)	Retrospective, observational	FR, US	7,156	Temperature increase > 5 °C	1 study	1 study	N/A	
Parkinson’s disease	3 (3,6%)	Retrospective, observational	ES, FR, US	204,656	>31.7 °C in 3 studies	N/A	2 studies	1 study	
Epilepsy	1 (1,2%)	Retrospective, observational	DE	604	>20 °C	N/A	Decreased	N/A	
Neuroinfectious disorders (TBE)	4 (4.8%)	Retrospective, observational	CZ, SE, SI	N/A	N/A	4 studies	N/A	N/A	
Notes.

Abbreviations TBE Thick-Borne encephalitis

Temp temperature

N/A Not applicable or available

AU Australia

CA Canada

CN China

CZ Czech Republic

DE Germany

DK Denmark

ES Spain

FR France

GB United Kingdom

IN India

IS Israel

IT Italy

JP Japan

KR South Korea

PR Puerto Rico

QA Qatar

RU Russia

SE Sweden

SI Slovenia

TR Turkey

TW Taiwan

US United States

VN Vietnam

* Sum of percentage exceeds 100% as some of the studies report on effect of temperature on several diagnoses.

Table 2 Studies (n = 9) investigating the differences in neurological disorders between migrants and non-migrant populations from their countries of origin.

Article	Site	Study design	Recruitment	Population (participants, sex, age)	Country of origin → arrival	Effect on epidemiology (increase in incidence/ prevalence/ hospitalization)	Mortality	
Stroke	
Hayfron-Benjamin et al. (2019)	DE, GB, GH, NL	Observational	Retrospective	206 GH, 30,1%M, 52.86 ± 9.9y; 444 DE/GB/NL, 50%M, 52.2 ± 8.8y	GH → GB, DE, NL	Decreased	N/A	
Chiu et al. (2010)	CN, HK, NAm, SG, TW, WEu,	Observational	Prospective	680 CN, 69.6%M, 65.7 ± 9y; 1,648 HK/ SG/TW, 68%M, 65.5 ±9.8y; 169 WEu, 71%M, 67.3 ±9.1y; 441 NAm, 63.3%M, 69.8 ±10.4y	CN → HK/NAm/ SG/TW/WEu	Decreased	No Effect	
Wolfe et al. (2006)	BB, GB	Observational	Retrospective	665 BB, 42.4%M, 71.2 ± 14.9 271 GB, 66.1 ± 13.7y	BB → GB	Increased	Increased	
Multiple Sclerosis	
Guimond et al. (2014)	CA, IR	Observational	Retrospective	Onset of MS before migration: 29 IR, 20.1%M, 24 (19-31)y Onset of MS after migration 48 IR, 33.3%M, 34 (25-40)y	IR → CA	Increased	N/A	
Hammond, English & McLeod (2000)	AU, GB, IE	Observational	Retrospective	331 GB/IE, 208/331 with age 20-49y in 1981	GB, IE → AU	Decreased	N/A	
Merle et al. (2005)	Caribbean islands, MQ	Observational	Prospective	53 Afro Caribbean, 13.2%M, 40.7 ± 12.1y 59 MQ, 18.6%M, 43.2 ± 10.4y	Caribbean → MQ	Decreased	N/A	
Comini-Frota et al. (2013)	BR, IT, ES, PT	Observational	Retrospective	652 BR/ES/IT/PT, 28.4%M, 42y	ES, IT, PT → BR	No Effect	N/A	
Nielsen et al. (2019)	DK	Observational	Retrospective	1,176,419 1st generation immigrants 184,282 2nd generation immigrants 7,607,816 Etnich Danish	All immigrants → DK	Increased	N/A	
Alzheimer and non-Alzheimer dementia	
Yamada et al. (2002)	BR, JP	Observational	Prospective	157 JP, 44,6%M, 70-100y	JP → BR	Increased	Increased	
Notes.

Abbreviations N/A Not applicable or available

M males

MS multiple sclerosis

y years

AU Australia

BB Barbados

BR Brazil

CN China

CA Canada

DE Germany

DK Denmark

ES Spain

GB United Kingdom

GH Ghana

HK Hong Kong

IE Ireland

IR Iran

IT Italy

JP Japan

MQ Martinique

NAm North America

NL Netherlands

PT Portugal

SG Singapore

TW Taiwan

WEu Western Europe

Epilepsy

Only one epilepsy study by Rakers et al. (2017) met inclusion criteria, a case-crossover study with 604 patients. The results indicated a 46% lower risk of admissions for epileptic seizures one day after exposure to temperatures above 20 °C. Further, an inverse association was found between epileptic seizures and low atmospheric pressure and high humidity (Rakers et al., 2017).

Headache and migraine

Two studies from two countries including N = 7,156 cases reported on the effects of high temperatures on headache (33% migraines) (Mukamal et al., 2009; Neut et al., 2012). The largest study by Mukamal et al. analyzed data from 7,054 patients seen in an emergency department between 2000–2007 (Mukamal et al., 2009). In this study, higher ambient temperatures in the 24 h preceding hospital presentation increased the risk of acute headache requiring emergency evaluation with 7.5% for each 5 °C increment in temperature. Neut et al. interviewed 102 children and adolescents with migraine and/or their parents about triggering factors precipitating migraine attacks. 70% stated warm climate could trigger their migraine, and 24% reported warm climate was often or very often a trigger factor for migraine attacks (Neut et al., 2012).

Multiple sclerosis

Six studies from five countries including N = 5,305 addressed the effects of high temperatures on patients with MS (Ogawa, Mochizuki & Kanzaki, 2004; Simmons et al., 2004; Tataru et al., 2006; Leavitt et al., 2012; Roberg & Bruce, 2016; Stellmann et al., 2017). Four studies found high temperatures to be associated with worsening of symptoms in these patients (Ogawa, Mochizuki & Kanzaki, 2004; Simmons et al., 2004; Leavitt et al., 2012; Stellmann et al., 2017). Stellmann et al. (2017) included 1,254 MS patients and showed that mobility deficits as measured by the ‘timed 25 foot walking test’ were increased so that every 10 °C increase of ambient temperature prolonged walking time by 0,35 s. In an anonymized self-reported survey by Simmons et al. (2004) 70% of multiple sclerosis participants reported worsening of their symptoms with high temperatures. Poor cognitive performance significantly correlated with higher outdoor temperatures in another group of 40 multiple sclerosis patients (Leavitt et al., 2012). Furthermore, the latter study also found a significant correlation between poor cognitive performance and high temperatures over a period of 6 months, suggesting a negative impact of warm temperatures on cognitive function in MS patients both when analyzed cross-sectionally and longitudinally (Leavitt et al., 2012). Higher frequencies of MS attacks were recorded during the warmest months in a small sample of 34 patients (Ogawa, Mochizuki & Kanzaki, 2004). The remaining two studies did not show any effects of high temperatures on MS symptomatology (Tataru et al., 2006; Roberg & Bruce, 2016). Thus, majority of the studies included (4 of 6 studies) reported progression of symptoms in MS patients during warm periods.

Parkinson’s disease

Three studies from three countries (FR, ES and US) with a total N = 204,656 evaluated the effects of high temperatures on Parkinson’s disease (Zanobetti et al., 2013; Pathak et al., 2005; Linares et al., 2016). Results from one study indicated a correlation between high ambient temperature (>34 °C) and an increase risk of excess morbidity and mortality in 3,287 PD patients (Linares et al., 2016). In a small French study with 36 PD patients, there was a trend towards more frequent autonomic failure during heat waves (Pathak et al., 2005). In contrast, Zanobetti et al. investigated the effects of extreme hot days (maximum temperature of 31,7 °C) on mortality in 201,333 patients with a diagnosis of PD but found no association between mortality and ambient temperatures (Zanobetti et al., 2013).

Stroke, overview

In total, 58 papers dealing with the effects of ambient temperatures on cerebrovascular morbidity and mortality were included (Zanobetti et al., 2013; Dawson et al., 2008; Feigin et al., 2000; Stafoggia et al., 2008; Vaneckova et al., 2008; Basu & Ostro, 2008; Knowlton et al., 2009; Lin et al., 2009; Wang et al., 2009; Green et al., 2010; Huang, Kan & Kovats, 2010; Basagaña et al., 2011; Wichmann et al., 2011b; Wichmann et al., 2011a; Basu et al., 2012; Cowperthwaite & Burnett, 2011; Ha et al., 2014; Lim, Kim & Hong, 2013; Shaposhnikov et al., 2014; Vaneckova & Bambrick, 2013; Wang, Gao & Wang, 2013; Chen et al., 2017a; Chen et al., 2017b; Chen et al., 2013; Chen et al., 2018; Wang & Lin, 2014; Zhang et al., 2014; Méndez-Lázaro et al., 2016; Vodonos et al., 2016; Guo et al., 2017; Han et al., 2017; Ponjoan et al., 2017; Zhang et al., 2016a; Zhou et al., 2017; Bai et al., 2018; Fu et al., 2018; Luo et al., 2018; Sherbakov et al., 2018; Bao et al., 2019; Shimomura et al., 2019; Ostro et al., 2010; Tarnoki et al., 2017; Oudin, Strömberg & Jakobsson, 2010; Salam et al., 2019; Goggins et al., 2012; Mostofsky et al., 2014; Hori, Hashizume & Tsuda, 2012; Matsumoto, Ishikawa & Kajii, 2010; Hong et al., 2003; Myint et al., 2007; Zheng et al., 2016; Çevik et al., 2015; Kyobutungi et al., 2005; Bobb et al., 2014; Yang et al., 2018; Harlan et al., 2014; Qian et al., 2008; Yang et al., 2016). The studies were from 18 different countries with a total case population of N = 5,869,284. Ischemic stroke was reported in 1,812,457, hemorrhagic stroke in 446,407, and subarachnoid hemorrhage (SAH) in 7,160 cases. In the remaining 3,603,260 cases, stroke subtypes were not specified.

Stroke, hospitalizations

Associations between stroke-related hospitalization and ambient temperatures were reported in 38 studies. Thirteen studies reported increasing risks for hospitalizations for stroke with higher ambient temperatures (Dawson et al., 2008; Wang et al., 2009; Ha et al., 2014; Shaposhnikov et al., 2014; Chen et al., 2017b; Vodonos et al., 2016; Sherbakov et al., 2018; Bao et al., 2019; Shimomura et al., 2019; Ostro et al., 2010; Tarnoki et al., 2017; Oudin, Strömberg & Jakobsson, 2010; Salam et al., 2019). Maximum temperatures were 12.9 °C in one study (Dawson et al., 2008), 20–27 °C in five studies (Wang et al., 2009; Ha et al., 2014; Sherbakov et al., 2018; Tarnoki et al., 2017; Oudin, Strömberg & Jakobsson, 2010), and >30 °C in five studies (Vaneckova & Bambrick, 2013; Chen et al., 2017b; Vodonos et al., 2016; Bao et al., 2019; Salam et al., 2019). Temperatures were unspecified in two studies (Shimomura et al., 2019; Ostro et al., 2010). Chen et al. (2017a) found excess hospitalizations due to ischemic stroke with high temperatures, while the risk for admissions because of hemorrhagic stroke was increased during both high and low temperatures. Green et al. (2010) and Basu et al. (2012) also found increased risks of hospitalization due to ischemic stroke with high temperatures (25.3 and 30.1 °C, respectively), but a decreased risk for hemorrhagic stroke. Bai et al. (2018) reported an increase in risk of stroke admissions both with high and low ambient temperatures. In 10 studies, lower ambient temperatures (−16.94 to 17 °C) were associated with increased risk of stroke-related hospitalizations (Wang & Lin, 2014; Guo et al., 2017; Luo et al., 2018; Goggins et al., 2012; Mostofsky et al., 2014; Hori, Hashizume & Tsuda, 2012; Matsumoto, Ishikawa & Kajii, 2010; Hong et al., 2003; Myint et al., 2007; Zheng et al., 2016). In the study by Wang, Gao & Wang (2013) there was an inverse correlation between temperatures and ischemic stroke admissions, leading to an increase of admissions during cold spells and a decrease during heat waves. Lin et al. (2009) reported lower risks of stroke admissions associated with high ambient temperatures (31.7 °C), even though the results were not statistically significant. Çevik et al. (2015) only found association between lower temperature and increased risk of SAH, and as in the remaining eight studies no associations between ambient temperatures and stroke-related hospitalizations was found (Feigin et al., 2000; Knowlton et al., 2009; Wichmann et al., 2011a; Cowperthwaite & Burnett, 2011; Vaneckova & Bambrick, 2013; Ponjoan et al., 2017; Kyobutungi et al., 2005; Bobb et al., 2014). Overall, most studies (17 of 24) reported higher risk of stroke-related hospitalizations during periods of high ambient temperatures.

Stroke, mortality

Twenty-one studies reported on the association between ambient temperatures and cerebrovascular mortality (Zanobetti et al., 2013; Stafoggia et al., 2008; Vaneckova et al., 2008; Basu & Ostro, 2008; Huang, Kan & Kovats, 2010; Basagaña et al., 2011; Wichmann et al., 2011b; Lim, Kim & Hong, 2013; Chen et al., 2013; Chen et al., 2018; Zhang et al., 2014; Méndez-Lázaro et al., 2016; Han et al., 2017; Zhang et al., 2016a; Zhou et al., 2017; Fu et al., 2018; Myint et al., 2007; Yang et al., 2018; Harlan et al., 2014; Qian et al., 2008; Yang et al., 2016). In 11 of the 21 studies (Zanobetti et al., 2013; Stafoggia et al., 2008; Vaneckova et al., 2008; Huang, Kan & Kovats, 2010; Basagaña et al., 2011; Lim, Kim & Hong, 2013; Méndez-Lázaro et al., 2016; Zhou et al., 2017; Yang et al., 2018; Harlan et al., 2014; Qian et al., 2008), increased risk of cerebrovascular death was associated with high ambient temperatures, including very high temperatures (>30 °C). Five studies reported an increase in cerebrovascular mortality with both low and high temperatures (Chen et al., 2013; Chen et al., 2018; Han et al., 2017; Zhang et al., 2016a; Fu et al., 2018), while 2 studies found no effect (Basu & Ostro, 2008; Wichmann et al., 2011b). Zhang et al. (2014), Myint et al. (2007) and Yang et al. (2016) were the only three to report significantly higher risks of cerebrovascular death in periods with cold, but not warm temperatures. Overall, most studies (16 of the 21 studies) found a higher risk of cerebrovascular mortality with increasing ambient temperatures.

Tick-borne encephalitis

The effects of high ambient temperatures on tick-borne encephalitis were assessed in four studies from three countries (SE, CZ, SI) (Danielová et al., 2008; Lindgren & Gustafson, 2001; Lukan, Bullova & Petko, 2010; Zeman & Bene, 2004). Lindgren & Gustafson (2001) linked an increase in the incidence of tick-borne encephalitis in Sweden, starting in the mid-80s, to climate change with increasingly milder winters and earlier springs. Lukan, Bullova & Petko (2010) and Zeman & Bene (2004) analyzed respectively 1,786 and 8,700 cases of tick-borne encephalitis in the period of 1961–70 to 2004. During 1980 to 2004 they found an increase in tick-borne encephalitis foci in areas of increasing altitudes which corresponded to gradual rises in annual temperatures, indicating an effect of climate change and higher temperatures on local tick-borne encephalitis incidence. Overall, the studies indicated a rise in the total incidence of tick-borne encephalitis with milder temperatures in Sweden and increasing temperatures in higher altitude areas in the Czech Republic and Slovenia.

Neurological disorders in human migrant and refugee populations

Results from studies comparing migrants with populations from their country of origin are listed below and in Table 2.

Alzheimer’s and non-Alzheimer’s dementia

One observational study reported a prevalence of 12.1% of all type dementia in 157 elderly Japanese individuals (age >70 years) who had migrated from Okinawa, Japan, to Brazil (Yamada et al., 2002). In comparison, risk of all type dementia in Japanese individuals living in Okinawa was 7.3%. Prevalence of dementia in five other regions in Japan was between 5.2 to 11.8%. Overall, prevalence of all type dementia was higher in Japanese migrants in Brazil than in Japanese who had stayed in Japan.

Multiple Sclerosis

Five observational studies investigated the effects of migration on the frequency (Guimond et al., 2014; Hammond, English & McLeod, 2000; Nielsen et al., 2019) and morbidity (Merle et al., 2005; Comini-Frota et al., 2013) of MS. In the study by Guimond et al., MS prevalence was higher in migrants who came from Iran to British Columbia (Yamada et al., 2002), whereas in another study the incidence was higher in the population from the home country (UK and Ireland) compared to the incidence in a migrant population settling in Australia (Hammond, English & McLeod, 2000). In one large retrospective study from Denmark (considered a high risk country for MS with a prevalence of >60 pr. 100,000) including more than 9 million individuals between 1968 and 2015 from the Danish Civil Registration System, higher risk of MS in 1st generation immigrants was found compared to the official risk rates of MS in their birth countries, especially if migration was from low- and intermediate risk countries (e.g., MS prevalence of <5 pr. 100,000 and 5–60 pr. 100,000 respectively) (Nielsen et al., 2019). The risk of MS was higher the younger the age at time of migration, while risk of MS was unchanged in people immigrating at age 35 years or above. MS risk among second-generation immigrants whose parents were born in low risk areas was doubled (Nielsen et al., 2019). Merle et al. investigated visual impairment in MS patients from Martinique who had or had not migrated to metropolitan France for at least 1 year before the age of 15 years and found that MS symptoms were more frequent and severe in the non-migrant group (Merle et al., 2005). Yet Comini-Frota et al. (2013) found no difference of MS morbidity between an Italian migrant population in Brazil and non-migrant Italians.

Stroke

Three observational studies compared data on cerebrovascular disease in migrant populations with data from non-migrant populations (Hayfron-Benjamin et al., 2019; Chiu et al., 2010; Wolfe et al., 2006). Two studies reported a higher prevalence of stroke in non-migrant groups living in China (Chiu et al., 2010) and Ghana (Hayfron-Benjamin et al., 2019), as opposed to migrant populations living in Hong Kong, Singapore, Taiwan, Western Europe, and North America. After adjusting for conventional cardiovascular risk factors, results were no longer significant in one study (Hayfron-Benjamin et al., 2019). In contrast, Wolfe et al. observed a higher incidence of stroke in migrants from Barbados moving to South London, including an increased incidence for specific stroke subtypes such as total anterior cerebral infarctions, posterior cerebral infarctions and SAH (Wolfe et al., 2006).

Discussion

Although the United Nations has declared climate change the greatest challenge of the 21st century and although neurological disorders comprise the greatest share of DALYs globally, we were unable to identify a single adequately designed study addressing how climate change and its consequences will alter neurological practice in the future. This is concerning given that the evidence that we did find indeed suggests many ways by which the effects of global warming and human migration on neurological practice may unfold.

Ambient temperatures and neurological disorders

Most stroke studies showed a relationship between increasing ambient temperatures and higher rates of hospitalization and mortality. Heat exposure may be associated with hemoconcentration and hyperviscosity, impaired endothelial function and hemodynamic disturbances, including cardiac arrhythmia, thereby increasing risks for both ischemic and hemorrhagic stroke (Lavados, Olavarría & Hoffmeister, 2018).

Likewise, higher temperatures were reported to worsen cognitive symptoms such as agitation, hallucinations, irritability, sleep disturbances, anxiety and depression among others, and to increase mortality in patients with dementia. The pathogenesis is unclear, but impaired physiological adjustment to rising temperatures because of dysautonomia are known from patients with fronto-temporal dementia and Alzheimer’s disease and may cause dehydration, cardiorespiratory distress and susceptibility to drug side effects (Fletcher et al., 2015). Autonomic failure aggravated by high temperatures is also a plausible complication in Parkinson’s disease that may lead to orthostatic hypotension and ensuing risk for trauma from syncope and falls (Pathak et al., 2005).

With multiple sclerosis, most studies revealed a decline in motor and cognitive functions following a rise in ambient temperatures. While this is consistent with Uhthoff’s phenomenon, i.e., the well-known reversible aggravation of multiple sclerosis symptoms caused by the blocking or slowing of nerve conduction with heat (Opara et al., 2016), no studies have specifically addressed the importance of Uhthoffs’s phenomenon relative to attack rates, inflammatory mechanisms and secondary neurodegeneration with rising temperatures. Furthermore, high temperatures are trigger factors for headache and migraine (Mukamal et al., 2009; Neut et al., 2012). Heat is known to cause vasodilation which may contribute to vascular migraine (Goadsby, 2009; Shevel, 2011), and water deprivation can provoke secondary headaches including migraine (Blau, 2005).

As for epilepsy, there are well-established relationships with temperature and pediatric febrile seizures, seizures in conjunction with heat strokes (Leon & Bouchama, 2015) and Dravet syndrome-related seizures associated with hot water (Hattori et al., 2008). Furthermore, increased body temperature may trigger hippocampal neuronal activity in mesial temporal lobe epilepsy (Wieser, 2004). We only found one study (Rakers et al., 2017) of the effects of ambient temperature on epileptic seizures, which showed lower rates of admission due to epileptic seizures one day after exposure to temperatures above 20 °C. This finding contradicts other data on epileptic seizures in patients with epilepsy (Leon & Bouchama, 2015; Hattori et al., 2008; Wieser, 2004). In this study (Rakers et al., 2017), more than half of the population did not receive treatment with an antiepileptic drug (AED), and only <10% were treated with 2 or more AEDs, thus patients with moderate or severe epilepsy might have been underrepresented. Many epilepsy patients with frequent epileptic seizures are not admitted to the emergency room when having self-limiting seizures. Thus, the effects of high temperature on seizure frequency might be underestimated relying on data from an in-hospital epilepsy population.

In addition, using tick-borne encephalitis as a model disease for neuroinfections, we found evidence that vector-borne diseases are prone to spread from endemic areas to currently non-endemic regions with increasing humidity and rising temperatures (Danielová et al., 2008; Lindgren & Gustafson, 2001; Lukan, Bullova & Petko, 2010; Zeman & Bene, 2004).

Human migration and neurological disorders

The impact of migration on the prevalence, incidence and severity of major neurological disorders is substantial but not uniform, with limited data comparing migrants to non-migrant individuals in their home country. Social, economic and cultural characteristics, particularly access to health care services, of both the countries of origin and arrival influence results.

The higher prevalence of stroke in populations from mainland China compared to Chinese migrants in Western countries (Chiu et al., 2010) may be due to higher dietary salt intake (Zhao et al., 2004); poorly controlled hypertension (Gong & Zhao, 2016; Huang et al., 2019) and less accessible healthcare (Chiu et al., 2010; Song et al., 2018). Furthermore, the higher stroke prevalence in Black Caribbean’s from South London might rely on socioeconomic factors and lifestyle changes that could unmask genetic susceptibilities (Wolfe et al., 2006; Smeeton et al., 2009), as Black Caribbean immigrants are more prone to hypertension and diabetes (Agyemang et al., 2014; Bidulescu et al., 2015; Lane, Beevers & Lip, 2002).

Similarly, environmental and genetic factors impact the prevalence and morbidity of MS in both first- and second-generation immigrants (Nielsen et al., 2019; Wändell et al., 2020). With migration to higher risk countries such as Denmark and Sweden, prevalence of MS increases in the migrating population (Guimond et al., 2014; Nielsen et al., 2019; Wändell et al., 2020), whereas with migration from high to low risk countries, prevalence decreases (Hammond, English & McLeod, 2000). The increase in MS prevalence is higher the younger the immigrants are at time of migration (Nielsen et al., 2019). Genetic factors affect MS rates in the long term, i.e., in later generations, notably with massive immigration (Comini-Frota et al., 2013; Sloka, Pryse-Phillips & Stefanelli, 2005).

In addition, it must be borne in mind that the psychological trauma of being displaced, lack of employment, low socioeconomic status, poor housing conditions, social isolation and ethnic discrimination may have limited direct impact on neurological disease; but all these factors can worsen mood disorders and other psychiatric conditions and may prevent access to health care, which may have secondary effects on neurological morbidity and mortality (Virupaksha, Kumar & Nirmala, 2014; Selten et al., 2020; Selten, Van der Ven & Termorshuizen, 2020).

Current limitations and future directions

Heterogeneity related to study design, exposures, outcome measures, effect modifiers and data presentation limit comparison of study results. Further most studies were of low quality, based on retrospective data and prone to high level of selection bias. Method of measuring ambient temperatures (e.g., outdoor versus indoor) was not standardized, and the ranges of temperatures investigated were wide. Most studies did not account for factors such as humidity, air pollution and geography which may influence outcome measures. A major limitation is that the findings mainly relate to high-income populations, and that reliable data from low-income vulnerable populations (which are prone to become most severely impacted by climate change) (Climate Change, 2020; Climate change and disaster displacement, 2020) are lacking. In addition, studies comparing migrants to non-migrants from their country of departure were few and most did not distinguish between first- and second-generation immigrants and did not account for socioeconomical factors (i.e., high-income vs. middle-income vs. low-income populations). Finally, results regarding the effects of global warming on neurological health among migrants are based on very low numbers compared to global figures on international migration (World migration report, 2020).

Importantly, the effect sizes related to global warming and human migration, i.e., exactly how much these factors will influence the morbidity and mortality of neurological disorders, are entirely unknown, as is the influence of other factors associated with climate change such as loss of biodiversity, rising sea levels and drought (Fig. 2).

Figure 2 Schematic overview on how climate change might soon impact neurological practice.

Global warming and human migration (left, yellow) were covered in this review. Although we identified no studies addressing precisely how and to what extent rising environmental temperatures may affect neurological disorders and only few studies that investigated neurological disorders in human migrant populations, it seems reasonable to assume that both global warming and climate refugees will alter clinical practice of various neurological disorders, owing to alterations in prevalence, incidence, mortality, morbidity and disease semiology. However, global warming and human migration are only two aspects of climate change. Other factors (right, red) that may change neurological practice directly or indirectly and that were not addressed in this review include drought, rising sea levels and loss of biodiversity (here, a dead nurse shark). These factors might lead to altered neurological practice owing to effects related to food shortage, water insecurity and displacement of communities, as well as an increase in vector-borne diseases (here, a Tsetse fly and an Anopheles mosquito which are the vectors for African trypanosomiasis and cerebral malaria, respectively). Figure created with Biorender.com.

In this review we focused on the impact of changes in ambient temperatures on neurological disease burden, but humanitarian emergencies and global health consequences are likely to have even much more complex effects on neurological practice in the foreseeable future (Mateen, 2010).

Taking the lack of valid data into account, it is not surprising that, so far, predictions of the future epidemiology of neurological disorders are typically based on population growth and life expectancy but completely ignore the impact of climate change and its consequences (Dorsey et al., 2007; Wafa et al., 2020).

Conclusions

Although results were inconsistent and limited due to the heterogeneity of data, including high risk of bias and lack of reliable data from low-income countries, our review suggests that climate change will soon change neurological practice because it affects morbidity and mortality of all major neurological disorders. Adequately designed studies to start addressing this issue are urgently needed, including increased focus on low-income populations, which will require coordinated efforts from the entire neurological community.

Supplemental Information

Supplemental Information 1 PRISMA checklist

Click here for additional data file.

Supplemental Information 2 Rationale and contribution

Click here for additional data file.

Supplemental Information 3 Literature search and search strings

Click here for additional data file.

Additional Information and Declarations

Competing Interests

Author Contributions

Data Availability

Daniel Kondziella is an Associate Editor for Acta Neurologica Scandinavica and has received financial compensation for this from the publisher Wiley. The other authors declare that they have no conflict of interest related to the content of this article.

Moshgan Amiri and Costanza Peinkhofer conceived and designed the experiments, performed the experiments, analyzed the data, prepared figures and/or tables, authored or reviewed drafts of the paper, and approved the final draft.

Marwan H. Othman, Teodoro De Vecchi and Vardan Nersesjan performed the experiments, analyzed the data, authored or reviewed drafts of the paper, and approved the final draft.

Daniel Kondziella conceived and designed the experiments, analyzed the data, prepared figures and/or tables, authored or reviewed drafts of the paper, and approved the final draft.

The following information was supplied regarding data availability:

The raw data is available in the Supplemental File.

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
