# Peer review of "Global warming and neurological practice: systematic review"

_PeerJ, doi:10.7717/peerj.11941_

## Round 0.1 · original submission · Major Revisions

· Academic Editor

Major Revisions

Dear Author, Major revisions must be done for this manuscript.Thank You.

·

Basic reporting

The paper is Clear and unambiguous, professional English used throughout.
Only minor English grammatical error noted
Line 371 - were did not ----- to delete were
Line 400 Also, to change to In addition,

Experimental design

Research question well defined, relevant & meaningful. It is stated how research fills an identified knowledge gap.
Methods described with sufficient detail & information to replicate

Validity of the findings

Conclusions are well stated, linked to original research question & limited to supporting results.

Additional comments

Weldone. Thank you for the wonderful effort to look into the impact of temperature of climate to various neurological illness. Future directions or discussion in subsequent papers can be look into the effect of temperature to mitochondrial function. Possible explanation for the outcome could be related to mitochondrial dysfunction following climate change.

Reviewer 2 ·

Basic reporting

The Introduction would benefit from review of the major points relating to the subject at hand, which can be found in Reis et al. Climate change: What neurologists should know. In Chopra J, Sawhney MS, Neurology in Tropics, 2nd ed, Elsevier, 2015, pp. 923-932. The Introduction should include a summary of the influences of temperature change (re global warming) and poor nutrition (re migrants) on the human nervous system. The discussion on MS would benefit from a brief summary of the established differential geographic distribution of this disease (see also: https://neurosciencenews.com/climate-change-brain-15806/).

Experimental design

Experimental design and data validity is adequate except that “quantitative statistical analysis” was not employed, which stands in contrast to the major conclusion that significant heterogeneity exists across studies with respect to methodology, outcome measures, confounders and study design.

Validity of the findings

The main objective of the study is to identify how ambient temperatures influence the epidemiology and clinical aspects of major [groups of] neurological disorders. The second stated objective is to investigate the prevalence and incidence of neurological disorders in human refugee and migrant populations. To the extent that the authors investigated certain disorders (Alzheimer’s and non-Alzheimer’s dementia, epilepsy, headache/migraine, multiple sclerosis, Parkinson’s disease, stroke, and tick-borne encephalitis), the review is adequate.

Acceptable, but see below.

Additional comments

There are four major reservations: (a) there is no statement of the country/continent of origin of the studies (except for tick-borne encephalitis); (b) whether the studies analyzed are generalizable because they are/are not representative of high-income, middle-income and/or low-income populations; (c) why nutritional disorders are omitted given that both climate change and migration will impact diet; for example,(i) greater dependence in sub-Saharan Africa and South Asia on environmentally tolerant plants that harbor toxins with neurotoxic potential and (ii) increased contamination of seafood from algal blooms that are increasing worldwide in association with global warming; and (d) why neuroinfectious disorders are limited to tick-borne encephalitis when data for more important disorders that involve the nervous system, such as malaria, must exist.

English usage is correct, with some exceptions: “United Nations has” (not “United Nations have”; “data are” (not “data is”) , and “Japanese migrants” (not “migrated Japanese”).

---

## Round 0.2 · Minor Revisions

· Academic Editor

Minor Revisions

Dear Author,

Please revise your manuscript accordingly as per the comments of the reviewer.

Reviewer 2 ·

Basic reporting

The authors have made some edits but more work is required before this manuscript meets an acceptable standard.

The major issue is the limitation in the applicability of the findings to high-income countries. There is obvious and unfortunate irony in recognizing that global health is and will continue to be impacted by global warming in commencing an analysis of the subject that essentially is limited to globally non-representative high-income populations: (https://www.pewresearch.org/global/2015/07/08/mapping-the-global-population-how-many-live-on-how-much-and-where/) Similarly, neurological health among tiny numbers of international migrants is analyzed relative to global estimates of the number of international migrants, which amounted to 3.5% of the world’s population in 2019, with annual estimated growth rates of 3-4% [IOM World Migration Report 2020]. To a large extent, the authors will have been forced down these narrow pathways by the equally limited scope of published studies, but failure to recognize these limitations is very disappointing. At a minimum, these study limitations, plus the fact that the present findings are relevant to neurology in high-income countries, should be recognized in the Abstract, Discussion, and Conclusion.

The authors have added a reference (#6) and summarized in the content thus: ”The effects of environmental factors on neurological practice are multifaceted and complex, involving climate, temperature, biodiversity, toxins and differences in human habitation and many others.” Food availability, quality and nutritional status should be added. Note that the authors “excluded commentaries, summaries, reviews, editorials, animal studies, toxicological studies,…

Several small edits are still needed. Prominent among these are (a) imprecise statements such as “worsen” and (b) use of “hot”, “warm” and “warmer” to define “temperature”, which should be with defined stated limits as: “high”, “higher”, “low”, “lower”, etc.

Lines
213 deficits measured by a walking test slightly worsened [meaning?]
353 worsen cognitive symptoms [meaning?]
215 reported worsening of their symptoms [meaning?]
223 worsening of symptoms in MS patients during warm periods. [meaning?]
180, 365, 489 hot temperature [see above]
313 9 mio [?]
318 the younger [the] age
340 Although the United Nations have [has]
288, 353 warmer temperatures [see above]
365 hot tmperatures [see above]
360 As to [for] epilepsy
29, 275, 380 warm temperature [see above]
418 “Most studies did not account for factors that may influence temperatures such as humidity, air pollution and geography.” Is this correct? Certainly, these factors influence air quality and oxygen content, but do they “influence ambient temperature”.
Reference citations employ style differences in use of initial upper/lower case in titles: e.g. 123 cf 124; 117 cf 118, and 108 cf 109

Experimental design

See above

Validity of the findings

See above

Additional comments

see above

---

## Round 0.3 · accepted · Accept

· Academic Editor

Accept

I am happy to inform you that your revised manuscript has been accepted. Thanking you.

Reviewer 2 ·

Basic reporting

Acceptable

Experimental design

Acceptable

Validity of the findings

Acceptable